# Resource Allocation for Reconfigurable Intelligent Surface Assisted Dual Connectivity

**DOI:** 10.3390/s22155755

**Published:** 2022-08-01

**Authors:** Yoghitha Ramamoorthi, Masashi Iwabuchi, Tomoki Murakami, Tomoaki Ogawa, Yasushi Takatori

**Affiliations:** NTT Access Network Service Systems Laboratories, Nippon Telegraph and Telephone Corporation, Yokosuka 2390847, Japan; masashi.iwabuchi.vs@hco.ntt.co.jp (M.I.); tomoki.murakami.nm@hco.ntt.co.jp (T.M.); tomoaki.ogawa.yg@hco.ntt.co.jp (T.O.); yasushi.takatori.rk@hco.ntt.co.jp (Y.T.)

**Keywords:** base station, coverage, dual connectivity (DC), fairness, reconfigurable intelligent surface (RIS), resource allocation, RIS-assisted DC, throughput, user scheduling

## Abstract

The next generation 6G wireless systems are envisioned to have higher reliability and capacity than the existing cellular systems. The reconfigurable intelligent surfaces (RISs)-assisted wireless networks are one of the promising solutions to control the wireless channel by altering the electromagnetic properties of the signal. The dual connectivity (DC) increases the per-user throughput by utilizing radio resources from two different base stations. In this work, we propose the RIS-assisted DC system to improve the per-user throughput of the users by utilizing resources from two base stations (BSs) in proximity via different RISs. Given an α-fair utility function, the joint resource allocation and the user scheduling of a RIS-assisted DC system is formulated as an optimization problem and the optimal user scheduling time fraction is derived. A heuristic is proposed to solve the formulated optimization problem with the derived optimal user scheduling time fractions. Exhaustive simulation results for coverage and throughput of the RIS-assisted DC system are presented with varying user, BS, blockage, and RIS densities for different fairness values. Further, we show that the proposed RIS-assisted DC system provides significant throughput gain of 52% and 48% in certain scenarios when compared to the existing benchmark and DC systems.

## 1. Introduction

The increased requirements of network capacity and user data rate have been driving the 5G networks. The evolution of beyond 5G and the next generation 6G networks includes a smart radio environment that can be achieved by controlling the wireless channel in a specified manner. The reconfigurable intelligent surface (RIS) is a promising solution to control the wireless channel by altering electromagnetic properties of the signal [1]. The RIS is also widely known as an intelligent reflecting surface as it consists of smaller antenna elements capable of reflecting the information signal with particular phase. The phase and the reflection of the signal are actively controlled by the RIS. This capability makes the wireless channel controllable. Further, the RIS-assisted wireless communication aids in improving the coverage area. The millimeter wave (mmWave)-based cellular networks with high bandwidth are prone to high attenuation and pathloss. This property of mmWave networks makes it impossible to reach for long distance particularly if the user is in non line of sight (NLOS). Leveraging the advantage of RIS to control the channel in mmWave based cellular network is expected to improve the coverage and throughput of the users in high attenuation regions. Although the RIS-assisted wireless communication is in evolving stages, the appropriate resource allocation and scheduling schemes are still in the basic stages and need thorough investigation.

The different analytical free-space path loss models based on the electromagnetic properties of RISs have been characterized in [2] for RIS-assisted wireless communication systems. A comprehensive analysis of main operating principles of RISs and the current research challenges of these RISs has been discussed in detail in [3]. The far-field path loss model using optical techniques has been derived by authors in [4]. Further, they also explained how the small elements of the surface can be advantageous when jointly beamforming the signal in the desired direction. Two types of implementation of RIS based on reflect arrays or metasurfaces, and their prospective channel modeling and path loss models, have been discussed in [5]. The joint optimization of coordinated transmit beamforming at BS and the reflective beamforming vector at the RIS by maximizing signal-to-interference- plus-noise ratio (SINR) at the users, subject to transmit power constraints, have been considered in [6].

The RIS with multicell communication, in order to avoid inter cell interference of the cell-edge users, has been proposed in [7]. This inter cell interference has been eliminated by destructively combining the reflected signal from RIS. The multiple RISs aided wireless communication system, where multiple RISs jointly assist the link between BS and cell edge users, has been investigated in [8]. The joint RIS deployment and the design of reflection coefficients have been jointly considered in [9]. Some of the multiple access schemes, such as non orthogonal multiple access (NOMA), frequency division multiple access, and time division multiple access (TDMA) have also been considered during joint optimization in [9]. The joint optimization problem of channel, resource allocation, power allocation, and the reflection coefficients for a downlink (DL) RIS based NOMA has been investigated by authors in [10].

The RIS-aided coordinated multipoint (CoMP) with joint processing has been proposed and discussed in [11] where the deployed RIS serves the cell-edge users by utilizing the signal from multiple BSs in proximity. The resource allocation for RIS aided vehicular communications based on slowly varying large-scale fading channel has been studied in [12]. The optimization of rate, energy efficiency, and their trade-off in a RIS based system along with overhead estimation has been presented in [13]. The optimization of RIS passive beamforming and orthogonal frequency division multiple access (OFDMA) in a RIS based system has been proposed in [14]. A novel approach to enhance the achievable rate of a ODFM system using RIS has been presented in [15]. The detailed investigation of RIS-associated NOMA and orthogonal multiple access (OMA) schemes have been studied in [16]. A comprehensive comparison between RIS and relay based systems over realistic mmWave channels has been presented in [17].

Dual connectivity (DC) has been introduced in [18] for cellular networks to improve per-user rate. The user in DC utilizes radio resources from two BSs and serve the users. There are two BSs involved in DC. One is the primary BS (PBS) where the user is associated based on its maximum received power. The other one is secondary BS (SBS) from where the user receives its second maximum SINR. The eligible DC user is able to receive different traffic from these two BSs. The first step of this to ensure the amount of traffic routed from OBS to the user via SBS. The downlink traffic scheduling and the SBS traffic routing to SBS in DC has been derived and presented in [19]. The performance comparison between CoMP and DC based system for different users and BS density scenarios have been presented in [20].

RIS-assisted wireless communication requires significant attention when it comes to the multi cell connectivity. The multi connectivity with multiple RIS-assisted CoMP has been evaluated in the literature [11]. The different scheduling and traffic routing schemes of DC have also have been analyzed in [19,20]. The RIS-assisted CoMP architectures are expected to improve coverage if the two RIS or BSs are in proximity. The CoMP in general requires resources to be reserved for the CoMP users that give rise to coverage throughput trade-off [21]. Most of the literature considers either RIS-aided wireless systems or DC systems separately. To the best of our knowledge, this is the first work that proposes RIS-assisted DC where the eligible user is expected to utilize radio resources from two BSs via two RIS deployed in proximity. Based on the above motivation, the following contributions are made as follows.

The RIS-assisted DC architecture is proposed, which utilizes radio resources from two different BSs via two different RISs.The joint resource allocation of RIS-assisted DC is formulated as an optimization problem and the optimal user scheduling time fraction for RIS-assisted DC systems for an α-fair scheduler is derived.The formulated joint problem is decomposed and the optimal user scheduling time fraction is derived using Karush–Kuhn–Tucker (KKT) conditions. A heuristic for solving the overall optimization problem with the derived scheduling time fraction is also presented.Exhaustive simulation results are presented for distributed RIS architecture with varying densities of users, BSs, and blockage densities are presented.

The paper is organized as follows. The system model with various parameters is presented in Section 2. The joint problem formulation of RIS-assisted DC is presented in Section 3. The heuristic for solving RIS-assisted DC problems is presented in Section 4. The evaluated numerical results are presented in Section 5. The conclusion and future work is given in Section 6. The system model is presented next.

## 2. System Model

We consider the OFDMA based cellular system. The overall system model of RIS-assisted DC is shown in the Figure 1. Let us denote the set of base stations (BSs) as J={1,2,3,…,J} and the set of users as U={1,2,3,…,U}. The set of subchannels in the system is denoted by C={1,2,3,…,C}. The set of RIS in the system is denoted as R={0,1,2,3,…,R}. Whenever RIS is denoted as 0, it is considered as representing a line-of-sight (LOS) direct link between BS and the user, i.e., there is no RIS between BS *j* and user *u*. We assume that the RIS has *N* elements in the panel to reflect and coherently combine and beamform the signal. The set of antenna elements in the RIS is denoted by N={1,2,3,…,N}. Every antenna element has its own reflection coefficient and phase shift. The mathematical notations used in this paper are tabulated in Table 1. The physical channel modeling is presented in next subsection.

### 2.1. Physical Channel Model

We consider the time division duplexing (TDD) based system and we focus on downlink (DL) in this work. The direct channel gain $h_{u,j}^{0}$ between the BS *j* and the user *u* is given as follows. We denote $x_{u,j}^{0}$ as the binary association variable that is 1, if the direct link exists between BS *j* and the user *u* and 0, if it does not. The association $x_{u,j}^{0}$ is based on the maximum received power and it is given as
(1)$$\begin{array}{c} x_{u,j}^{0}=\left\{\begin{array}{c} 1,\mathrm{if}\;j=\arg\max_{j}\left\{P_{j}h_{u,j}^{0}\right\},\\ 0,\mathrm{otherwise},\forall u\in U,\;\forall j\in J,\; \end{array}\right. \end{array}$$
where $P_{j}$ is the transmit power of BS *j* and $h_{u,j}^{0}$ is the direct LOS channel gain between BS *j* and the user *u* as expressed below.
(2)$$h_{u,j}^{0}=10^{-\left(\frac{PL\left(u,j\right)+\upsilon +\xi -G_{j}-G_{u}}{10}\right)}$$
where PL(u,j) is the pathloss between user *u* and BS *j*, υ is the small scale fading loss, ξ is the loss due to log-normal shadowing, $G_{j}$ is the transmit antenna gain, and $G_{u}$ is the gain of user antenna. The pathloss between *u* and *j* is determined by the appropriate radio access terminal. The direct link SINR of user *u* from the BS *j* based on [21] is given as follows.
(3)$$\omega_{u,j}^{0}=\frac{P_{j}^{c}h_{u,j}^{0}}{\sum_{j^{\prime }\in J\backslash j}P_{j^{\prime }}^{c}h_{u,j^{\prime }}^{0}+\sigma^{2}},\;\forall u\in U,\;$$
where, $\sum_{j^{\prime }\in B\backslash b}P_{j^{\prime }}^{0}h_{u,j^{\prime }}^{0}$ is the interference from the other BSs, $P_{j}$ is the transmit power of BS *j*, and $\sigma^{2}$ is the noise power. The link rate and the scheduling of the system is presented next.

### 2.2. Link Rate and Scheduling

We consider flat fading channels where all subchannels and channels over elements have similar channel gains. We also consider an adaptive modulation and coding scheme (MCS) as in [21] with the SINR as in (Equation 3). Let Γ(.) denote the spectral efficiency obtained from MCS. Given SINR as in (Equation 3) and spectral efficiency Γ(.) (bits/symbol), the direct link rate of user *u* from BS *j* is
(4)$$l_{u,j}^{0}=\frac{\Gamma \left(\omega_{u,j}^{0}\right)\;SC_{OFDM}\;SY_{OFDM}C}{T_{sc}},\;j\in j,\;c\in C,$$
where, $SC_{OFDM}$, $SY_{OFDM}$, and $T_{sc}$ represent the number of subcarriers per subchannel, number of symbols per subcarrier, and time duration of a subframe, respectively. Given the direct link rate $l_{u,r}^{0}$ as in (Equation 4), the net data rate of user *u* from BS *j* for a conventional direct link is given as follows.
(5)$$\lambda_{u}=\sum_{j\in J}x_{u,j}^{0}\delta_{u,j}^{0}l_{u,j}^{0},\forall u\in U,$$
where the user association $x_{u,j}^{0}$ is as in (Equation 1), link rate $l_{u,j}^{0}$ as in (Equation 4), and $\delta_{u,j}^{0}$ is the user scheduling time fraction for user *u* by BS *j*. The $\delta_{u,j}^{0}$ for the direct channel between BS *j* and user *u* is given as
(6)$$\delta_{u,j}^{0}=\frac{{\left(l_{u,j}^{0}\right)}^{\frac{1-\alpha }{\alpha }}}{\sum_{u\in U}x_{u,j}^{0}{\left(l_{u,j}^{0}\right)}^{\frac{1-\alpha }{\alpha }}},\;\forall j\in J,$$
where α is the fairness parameter. The channel modeling parameters explained in (Equation 1)–(Equation 6) are related to direct channel parameters between BS *j* and the user *u*. These does not involve the RIS as an intermediate node. The channel modeling and user scheduling parameters of RIS based links are explained in the next subsection.

### 2.3. RIS Channel Model and Scheduling

The physical channel model of RIS based system consists of two channel links. One is the channel between BS *j* and RIS *r* denoted as $f_{r,j}^{n}$ and the other one is between RIS *r* and user *u*, denoted as $g_{u}^{n,r}$. The superscript *n* implies the $n^{th}$ reflective element. The overall channel between user *u* and BS *b* via RIS *r* is the cascading of the above two channels along with the reflective parameter matrix of of RIS (Θ), i.e., $f_{r,j}^{n}\Theta g_{u}^{n,r}$. The SINR of user *u* from BS *j* via RIS *r* is denoted by ${\hat{\omega }}_{u,j}^{r}$ as given below
(7)$$\omega_{u,j}^{r}=\frac{P_{j}\left(h_{u,j}^{0}+f_{r,j}^{n}\Theta g_{u}^{n,r}\right)}{\sum_{j^{\prime }\in J\backslash j}P_{j^{\prime }}h_{u,j^{\prime }}^{0}+\sigma^{2}},\;\forall u\in U,\;\forall r\geq 1\;$$
where, $h_{u,j}^{0}$ is the direct channel gain between BS *j* and user *u*, Θ is the diagonal matrix containing phase shifts ($\theta_{r,n}$) and reflection coefficient values ($\beta_{r,n}$) of each element *n* of RIS *r*. The $\sum_{j^{\prime }\in B\backslash b}P_{j^{\prime }}^{0}h_{u,j^{\prime }}^{0}$ is the interference from the other BSs, $P_{j}$ is the transmit power of BS *j*, and $\sigma^{2}$ is the noise power. The interference from the other RIS is negligible because the received signal is assumed to be coherently combined and beamformed towards the intended receiver *u*. When the diagonal matrix is expanded in terms of ($\theta_{r,n}$) and ($\beta_{r,n}$), the SINR equation is expressed as
(8)$$\omega_{u,j}^{r}=\frac{P_{j}\left(h_{u,j}^{0}+\sum_{n=1}^{N}f_{j}^{r,n}\beta_{r,n}e^{j\theta_{r,n}}g_{u}^{r,n}\right)}{\sum_{j^{\prime }\in J\backslash j}P_{j^{\prime }}^{c}h_{u,j^{\prime }}^{0}+\sigma^{2}},\;\forall u\in U,\forall r\geq 1\;$$
where *N* is the total number of antenna elements in RIS, $\beta_{r,n}$, and $\theta_{r,n}$ is the reflection coefficient and phase shift of $n^{th}$ antenna element of $r^{th}$ RIS, respectively. Similar to (Equation 4), if the user *u* is associated with BS *j* via RIS *r*, then the link rate would be a function of $\omega_{u,j}^{r}$ and is given as follows.
(9)$$l_{u,j}^{r}=\frac{\Gamma \left(\omega_{u,j}^{r}\right)\;SC_{OFDM}\;SY_{OFDM}C}{T_{sc}},\;j\in j,\;c\in C\;.$$

Given the link rate *r* as in the above equation, the actual data rate of the user *u* in the RIS system is expressed as
(10)$$\lambda_{u}=\sum_{j\in J}\sum_{r\in R}x_{u,j}^{r}\delta_{u,j}^{r}l_{u,j}^{r},\forall u\in U,$$
where the user association $x_{u,j}^{0}$ as in (Equation 12), link rate $l_{u,j}^{r}$ as in (Equation 9), $\delta_{u,j}^{r}$ is the user scheduling time fraction for user *u* by BS *j* via RIS *r*. The $\delta_{u,j}^{r}$ from BS *j* to user *u* via RIS *r* is given as
(11)$$\delta_{u,j}^{r}=\frac{{\left(l_{u,j}^{r}\right)}^{\frac{1-\alpha }{\alpha }}}{\sum_{u\in U}\sum_{r\in R}x_{u,j}^{r}{\left(l_{u,j}^{r}\right)}^{\frac{1-\alpha }{\alpha }}},\;\forall j\in J.$$
where α is the fairness parameter, and $x_{u,j}^{r}$ is the user association between the BS *j* and user *u* via RIS *r*. This is expressed as follows.
(12)$$\begin{array}{c} x_{u,j}^{r}=\left\{\begin{array}{c} 1,\mathrm{if}\;j=\arg\max_{j,r}\left\{\omega_{u,j}^{r}\right\},\\ 0,\mathrm{otherwise},\forall u\in U,\;\forall j\in J,\;\forall r\in R.\; \end{array}\right. \end{array}$$
The performance metrics considered in RIS-assisted DC systems are considered next.

### 2.4. Performance Metrics

We consider coverage and α-fair throughput as our performance metric in this work. The coverage is defined as the probability of user receiving SINR greater than the minimum threshold as in [21]. We consider α-fairness in this work. The utility function of $\Lambda_{\alpha }\left(x\right)$ with respect to general variable *x* as in [21] is written as follows.
(13)$$\Lambda_{\alpha }\left(x\right)=\left\{\begin{array}{cc} \frac{x^{1-\alpha }}{1-\alpha }, & \alpha >0,\;\alpha \neq 1,\\ log(x) & \alpha =1. \end{array}\right.$$
where α is the fairness parameter. When α=1, (Equation 13) leads to proportional fairness. As α increases, the worst case user’s rate of the system improves. α→∞ corresponds to max-min fairness where then minimum rate of the user in the system is maximized. The optimization problem formulated for RIS-assisted DC systems is presented in the next section.

## 3. RIS Assisted Dual Connectivity

We consider $x_{u,j}^{r}$ as a binary association variable that indicates whether user *u* is associated with *j* via RIS *r* or not. The tuple (u,r,j) in all the variables indicates the identification values of user, RIS, and BSs. When the value of *r* is 0, then there is no RIS participating in the link between the user *u* and BS *j*. If *r* is not equal to 0, then the the user *u* is served by BS *j* via RIS *r*. We denote $d_{u,j}^{r}$ is the association variable between BS *j* and user *u* via RIS *r*. When $d_{u,j}^{r}=1$, then the user *u* receives SINR greater than the threshold τ from two different BS via two different BSs. The DC association variable of user *u* with BS *k* via RIS *q* denoted as $d_{u,k}^{q}$ is 1 only when its primary association $x_{u,k}^{r}$ is 1 with some other BS *j* via RIS *r*. Given the channel ($h_{u,j}^{0},f_{j}^{r,n},g_{u,j}^{r,n}$) and the SINR ($\omega_{u,j}^{r}$), the overall optimization problem for an α-fair utility function is presented as follows.
(14)$$\begin{array}{ccc} P1 & : & \max_{\begin{array}{c} \beta_{r,n},\theta_{r,n},\delta_{u,j}^{r},\\ \tau ,x_{u,j}^{r},d_{u,j}^{r} \end{array}}\sum_{u\in U}\Lambda_{\alpha }\left(\lambda_{u}\right), \end{array}$$
(15)$$\begin{array}{ccc} s.t\;\lambda_{u} & = & \sum_{r\in R}\sum_{j\in J}x_{u,j}^{r}\left(l_{u,j}^{r}\delta_{u,j}^{r}+\sum_{r\in R}\sum_{k\in J}d_{u,k}^{r}l_{u,k}^{r}\delta_{u,k}^{r}\right),\forall u\in U, \end{array}$$
(16)$$\begin{array}{ccc} & & \sum_{u\in U}\sum_{r\in R}\left(x_{u,j}^{r}+d_{u,j}^{r}\right)\delta_{u,j}^{r}\leq 1,\;\forall j\in J,\; \end{array}$$
(17)$$\begin{array}{ccc} & & \sum_{r\in R}\sum_{j\in J}\left(x_{u,j}^{r}+d_{u,j}^{r}\right)\leq 1,\;\forall u\in U,\\ & & x_{u,j}^{r}\;\mathrm{as}\;\mathrm{in}\;\left(12\right), \end{array}$$
(18)$$\begin{array}{ccc} d_{u,k}^{q} & = & \left\{\begin{array}{c} 1,\mathrm{if}\;k,q=\arg\max_{\begin{array}{c} k\in J\backslash \{j\}\\ q\in R\backslash \{r\} \end{array}}\left\{\omega_{u,k}^{q}\right\},\;\\ \;\;\;\;\;\;\;\omega_{u,j}^{r}\geq \tau ,\omega_{u,k}^{q}\geq \tau ,\\ 0,\mathrm{otherwise},\forall u\in U,\; \end{array}\right. \end{array}$$
(19)$$\begin{array}{ccc} & & x_{u,j}^{r}\in \left\{0,1\right\},\;\forall u\in U,\;\forall j\in J,\;\forall r\in R, \end{array}$$
(20)$$\begin{array}{ccc} & & d_{u,j}^{r}\in \left\{0,1\right\},\;\forall u\in U,\;\forall j\in J,\;\forall r\in R, \end{array}$$
(21)$$\begin{array}{ccc} & & \delta_{u,j}^{r}\geq 0,\;\forall u\in U,\;\forall j\in J,\;\forall r\in R,\\ & & \omega_{u,j}^{r}\;\mathrm{as}\;\mathrm{in}\;\left(8\right),l_{u,j}^{r}\;\mathrm{as}\;\mathrm{in}\;\left(9\right), \end{array}$$
(22)$$\begin{array}{ccc} & & \beta_{r,n}\in \left[0,1\right],\; \end{array}$$
(23)$$\begin{array}{ccc} & & \theta_{r,n}\in \left[0,\pi \right],\; \end{array}$$
where (Equation 14) is the maximization of the α-fair utility function defined in (Equation 13) with respect to RIS parameters amplitude reflection coefficient $\beta_{r,n}$, phase shift coefficient $\theta_{r,n}$, the user scheduling time fraction $\delta_{u,j}^{r}$, and the binary variables $x_{u,j}^{r}$, and $d_{u,j}^{r}$. The constraint in (Equation 15) specifies the actual data rate of the user *u*. If *u* has the association only with its primary BS (PBS) via the RIS *r*, then only $x_{u,j}^{r}$ is 1. However, if *u* is a DC user via RIS *r*, then $d_{u,j}^{r}$ is also equal to 1. Thus DC user’s data rate is computed from both the terms as in (Equation 15). However, if association is between the BS *j* and user *u* without RIS, then only $x_{u,j}^{0}$ is 1. This is because the user *u* already has the strong direct LOS channel from BS *j*. The constraint in (Equation 16) specifies that the sum of time fraction allocated to all the users by any BS *j* directly or via any RIS *r* can not exceed 1. Further, the constraint in (Equation 17) shows that the user *u* can be associated to one *j*, i.e., if it associated to primary BS (PBS) *j*, then $x_{u,j}^{r}=1$ or if it associates to the secondary BS (SBS) *j*, then $d_{u,j}^{r}=1$. Both $x_{u,j}^{r}$ and $d_{u,j}^{r}$ will not be 1 at the same time for the same BS *j* and RIS *r*. The condition for user association with PBS and its associated RIS is given in the constraint (Equation 12). The user *u* is associated to the PBS *j* via the RIS *r* (if r=0, without RIS) that has maximum SINR. Similarly, the condition for DC user association with its SBS *k* based on the constraint (Equation 18). The user is qualified to be a DC user if its SINR from both PBS ($\omega_{u,j}^{r}$) and SBS ($\omega_{u,k}^{q}$) via different RIS should greater than the threshold τ. The binary constraints of association variables are specified in (Equation 19) and (Equation 20), respectively. The positivity constraint on user scheduling time fraction through RIS *r* is given in (Equation 21). Since the set of RIS R also contains the value 0, this positivity constraint is ensured for the scheduling time fraction for the direct LOS link between BS *j* and user *u* also. The constraints on amplitude reflection coefficient $\beta_{r,n}$ and the phase shift $\theta_{r,n}$ of the $n^{th}$ antenna element of $r^{th}$ RIS are given in (Equation 22) and (Equation 23), respectively.

The optimization problem presented in (Equation 14) is mixed integer non linear programming (MINLP) and is difficult to solve for all variables at once. Therefore, (Equation 14) is decomposed into subproblems that can be solved independently. Given the channel, $f_{j}^{r,n}$ and $g_{u,j}^{r,n}$, the maximum rate is achieved by selecting $\theta_{r,n}=∠h_{u,j}^{0}-∠f_{j}^{r,n}g_{u,j}^{n}$ for each element *n* [22]. We assume a flat fading channel, i.e., the channel between BS *j* and every element *n* of the RIS *r* is the same. Therefore, the channel between BS *j* and RIS *r* is now denoted as $f_{j}^{r}$. Similarly, the channel between RIS *r* and user *u* as $g_{u,j}^{r}$ and the amplitude of the reflection coefficient is 1. The entire RIS is now focusing towards a single user *u*. The modified SINR with the channel and the optimal phase shifts can be conclusively written as in [22] as
(24)$$\omega_{u,j}^{r}=\frac{P_{j}{\left(\sqrt{h_{u,j}^{0}}+N\sqrt{f_{j}^{r}g_{u}^{r}}\right)}^{2}}{\sum_{j^{\prime }\in J\backslash j}P_{j^{\prime }}h_{u,j^{\prime }}^{0}+\sigma^{2}},\;\forall u\in U,\forall r\in R,r\geq 1\;\;$$
where $\sum_{j^{\prime }\in J\backslash j}P_{j^{\prime }}h_{u,j^{\prime }}^{0}$ is the interference received by the user *u* from other BSs, $P_{j}$ is the transmit power. Since the RIS parameters are optimized and the SINR is computed as in (Equation 24), the goal is now to determine the normal , RIS, and RIS-assisted DC users based on the computed SINR values. Given the SINR of *u* from BS *j* via RIS *r*, the link rate of the given link $l_{u,j}^{r}$ is computed as in (Equation 9), the global optimization problem is decomposed and formulated as user scheduling problem as below.
(25)$$\begin{array}{ccc} P2 & : & \max_{\delta_{u,j}^{r}}\sum_{u\in U}\Lambda_{\alpha }\left(\lambda_{u}\right)\\ & & s.t\;(15)-(21) \end{array}$$
the problem in P2 is the maximization of α-fair utility function given the association, SINR and the link rates. (Equation 25) is now a convex problem with respect to the user scheduling time fraction $\delta_{u,j}^{r}$ based on [21] and it is solved using KKT conditions. The proposition for deriving optimal user scheduling time fraction $\delta_{u,j}^{j}$ is presented next.

**Proposition** **1.**
*Given the binary association of user u with BS j with RIS (r=0) or without RIS (r>1), the secondary association $d_{u,j}^{r}$ based on the next maximum received SINR, the link rate of user u with BS j as a normal or DC user, the user scheduling time fraction allocated to the user u by BS j with or without RIS is given as follows.*

(26)
$$\delta_{u,j}^{r}=\frac{\left(x_{u,j}^{r}+d_{u,j}^{r}\right){\left(l_{u,j}^{r}\right)}^{\frac{1-\alpha }{\alpha }}}{\sum_{u\in U}\sum_{r\in R}\left(x_{u,j}^{r}+d_{u,j}^{r}\right){\left(l_{u,j}^{r}\right)}^{\frac{1-\alpha }{\alpha }}},\;\forall j\in J.$$



**Proof.** Given the binary user association for the tuple (u,j,r), the optimization problem in (Equation 14) is convex with respect to the variable $\delta_{u,j}^{r}$. The equivalent Lagrangian function of the (Equation 14) can be defined as follows.
(27)$$\begin{array}{ccc} L & = & -\sum_{u\in U}\frac{{\left(\sum_{r\in R}\sum_{j\in J}x_{u,j}^{r}l_{u,j}^{r}\delta_{u,j}^{r}+\sum_{r\in R}\sum_{k\in J}d_{u,k}^{r}l_{u,k}^{r}\delta_{u,k}^{r}\right)}^{1-\alpha }}{1-\alpha }+\sum_{j\in j}Y_{j}\left(\sum_{u\in U}\sum_{r\in R}\left(x_{u,j}^{r}+d_{u,j}^{r}\right)\delta_{u,j}^{r}-1\right)\\ & + & \sum_{j\in j}W_{j}\left(\sum_{u\in U}\sum_{r\in R}\left(x_{u,j}^{r}+d_{u,j}^{r}\right)-1\right)-\sum_{u\in U}\sum_{r\in R}\sum_{j\in J}Z_{u,j}^{r}\delta_{u,j}^{r} \end{array}$$
where the $Y_{j}$, $W_{j}$, and $Z_{u,j}^{r}$ are intermediate Lagrangian variables. The necessary KKT conditions to compute the optimal solution for the given problem are as follows.
(28)$$\begin{array}{ccc} \frac{dL}{d\delta_{u,j}^{r}} & = & 0, \end{array}$$
(29)$$\begin{array}{ccc} \sum_{j\in J}Y_{j}\left(\sum_{u\in U}\sum_{r\in R}\left(x_{u,j}^{r}+d_{u,j}^{r}\right)\delta_{u,j}^{r}-1\right) & = & 0, \end{array}$$
(30)$$\begin{array}{ccc} Z_{u,j}^{r}\delta_{u,j}^{r} & = & 0, \end{array}$$
where (Equation 28) is the first order necessary condition for optimal $\delta_{u,j}^{r}$. (Equation 29) and (Equation 30) are the complementary-slackness conditions. Applying (Equation 28) in (Equation 27), we get
(31)$$\begin{array}{c} -\left(x_{u,j}^{r}+d_{u,j}^{r}\right){\left(l_{u,j}^{r}\right)}^{1-\alpha }{\left(\delta_{u,j}^{r}\right)}^{-\alpha }+Y_{j}-Z_{u,j}^{r}=0, \end{array}$$
(32)$$\begin{array}{c} \left(x_{u,j}^{r}+d_{u,j}^{r}\right){\left(l_{u,j}^{r}\right)}^{1-\alpha }=Y_{j}-Z_{u,j}^{r}. \end{array}$$
The optimal $\delta_{u,j}^{r}$ should satisfy the positivity condition $\delta_{u,j}^{r}>0$ as given in (Equation 21) and should follow the complementary-slackness condition as in (Equation 30). Therefore, we consider $Z_{u,j}^{r}=0$. Using $Z_{u,j}^{r}=0$ in (Equation 32) and on further simplification,
(33)$$\begin{array}{c} \delta_{u,j}^{r}=\frac{\left(x_{u,j}^{r}+d_{u,j}^{r}\right){\left(l_{u,j}^{r}\right)}^{\frac{1-\alpha }{\alpha }}}{Y_{j}^{\frac{1}{\alpha }}}, \end{array}$$
substituting (Equation 33) in the constraint (Equation 19), we get,
(34)$$Y_{j}^{\frac{1}{\alpha }}=\sum_{u\in U}\sum_{r\in R}\left(x_{u,j}^{r}+d_{u,j}^{r}\right){\left(l_{u,j}^{r}\right)}^{\frac{1-\alpha }{\alpha }}$$Substituting (Equation 34) in the constraint (Equation 33), we get (Equation 26), and this completes the proof of Proposition 1.    □

Similarly, the optimal user scheduling time fraction can be derived for α=1 and α→∞. The user scheduling time fraction of user *u* from BS *j* via RIS *r* for proportionally fair (PF) scheduler with α=1 is given as
(35)$$\delta_{u,j}^{r}=\frac{\left(x_{u,j}^{r}+d_{u,j}^{r}\right)}{\sum_{u\in U}\sum_{r\in R}\left(x_{u,j}^{r}+d_{u,j}^{r}\right)},\;\forall j\in J.$$
Further, when α→∞, the user scheduler with max-min fairness for RIS-assisted DC system is written as follows.
(36)$$\delta_{u,j}^{r}=\frac{\left(x_{u,j}^{r}+d_{u,j}^{r}\right){\left(l_{u,j}^{r}\right)}^{-1}}{\sum_{u\in U}\sum_{r\in R}\left(x_{u,j}^{r}+d_{u,j}^{r}\right){\left(l_{u,j}^{r}\right)}^{-1}},\;\forall j\in J.$$
The heuristic for solving the given optimization problem with the derived optimal user scheduling time fraction is presented next.

## 4. Proposed Heuristic for RIS Assisted DC

In this section, we present the proposed heuristic for a RIS-assisted DC system as in Algorithm 1 for solving (Equation 14). The problem presented in (Equation 14) is the global optimization problem for the RIS-assisted DC system. This involves the selection of DC users, scheduling, and computation of its resultant data rate. Solving for all at once is NP-hard and is difficult to solve. We propose this heuristic for a step-by-step procedure to solve the optimization problem proposed in (Equation 14).
**Algorithm 1** Proposed heuristic for RIS-assisted DC1:INPUTS : $\left\{P_{j}^{r}h_{u,j}^{r}\right\}$, U, τ2:OUTPUT : $\lambda_{u}$3:Initialize : u = 1, $\{x_{u,j}^{r}\}=0$, $\{d_{u,j}^{r}\}=0$4:**Repeat**5:Sort $\{P_{j}^{r}h_{u,j}^{r}\},\forall r\geq 1$ in decreasing order of received power from RIS. The rearranged order is represented using *q*. r=0 implies direct link between BS and user.6:**if** 
$P_{j}h_{u,j}^{q}\geq P_{j}h_{u,j}^{0}$ 
**then**7:   $\omega_{u,j}^{q}=f\left(\left\{P_{j}h_{u,j}^{q}\right\}\right)$ as in (Equation 7)8:   $\omega_{u,k}^{q+1}=f\left(\left\{P_{k}h_{u,k}^{q+1}\right\}\right)$ as in (Equation 7)9:   $x_{u,j}^{q}=1$10:   **if** $\omega_{u,j}^{q}\geq \tau \;\;\;\&\;\;\;\omega_{u,k}^{q+1}\geq \tau$ **then**11:     $d_{u,k}^{q+1}=1$12:   **else**13:     $d_{u,k}^{q+1}=0$14:   **end if**15:**else**16:   $\omega_{u,j}^{0}=f\left(\left\{P_{j}h_{u,j}^{0}\right\}\right)$ as in (Equation 3)17:   $x_{u,j}^{0}=1$18:**end if**19:Set u=u+120:**Until** 
u>U21:Set u = 122:**Repeat**23:Compute ${\left(\delta_{u,j}^{r}\right)}^{*}$ as in (Equation 26)24:Compute $\lambda_{u}$ as in (Equation 15)25:Set u=u+126:**Until** 
u>U27:**Stop**

The set of received powers, user set, and the threshold for the selection of DC users (τ) are taken as inputs. The resultant data rate as in (Equation 15) is the output of the system. The heuristic is initialized by setting, u=1, binary association matrix $x_{u,j}^{r}=0$, and DC binary association matrix $d_{u,j}^{r}=0$. As an initial step, the received power of the user u=1 from all RIS is arranged in the decreasing order. The rearranged order of RIS based on the received power is now denoted with a variable *q*. The SINR of the user *u* is computed using (Equation 7) or (Equation 3) based on its received power. If the received power of users’ direct link with the BS is greater than the received power with RIS, then the SINR of the user *u* is computed using (Equation 3) and the corresponding $x_{u,j}^{0}$ is set to 1. If received power with RIS is greater than the direct link, then the SINR is computed using (Equation 7) and so the $x_{u,j}^{q}$ is set to 1. Further, we define DC users as eligible users to utilize radio resource from two different BSs via two different RIS. If the first two values of users’ received power in rearranged is greater than certain predefined threshold τ, then the user is considered to be RIS-assisted DC user. This user now utilizes radio from two different BSs via two different RISs. Given that all the SINR of users with direct and RIS link have been computed, the next step is to compute users’ resultant data rate ($\lambda_{u}$) as in (Equation 15). To compute $\lambda_{u}$, we need to first compute the optimal user scheduling time fraction ${\left(\delta_{u,j}^{r}\right)}^{*}$ as in (Equation 26). The complexity of the heuristic is in the order of 2U, where U is the total number of users in the system. The heuristic is divided into two main steps. The computation of SINR and the selection of normal, DC, and RIS-assisted DC users at the first step. The latter step continues with computation of optimal user scheduling time fraction and the resultant data rate. The numerical evaluation for different scenarios are presented next.

## 5. Numerical Results

We consider an area of 200 m × 200 m. We consider deterministic BSs and RIS locations. This is acceptable because the RIS is determined to be deployed based on the BS locations and its associated high blockage loss places. The users and blockers are deployed uniformly randomly in the considered area. The results of throughput are averaged over $10^{4}$ independent location realizations. For each location realization, fading is averaged over 100 realizations. The number of elements of RIS considered in this simulation is 40. The simulation parameters are listed in Table 2. The channel between BS and user is considered to be NLOS and its associated pathloss is computed from [23]. We evaluate the performance of the mmWave based cellular system. However, given the channel, the derived results are applicable for any system with any number of RIS elements in general. We consider evaluating the coverage and throughput performance of four systems, namely benchmark (Ben) as in [21], DC as in [20], RIS, and RIS-assisted DC (RISDC). The benchmark system is the conventional mmWave based cellular system without DC and RIS technologies. In DC, an eligible user is able to utilize radio resources from two different BSs directly. Since mmWave based cellular systems are prone to high blockage and attenuation, a controllable path with RIS has been considered. The eligible RIS users are served via RIS by the associated BSs. To further improve the per users’ rate, the proposed RISDC system leverage the advantage of DC in a RIS based system. In this RISDC system, the users eligible to be served as RISDC are identified and then RIS based users are also selected based on the predetermined SINR threshold. The RISDC system consists of both RISDC and RIS users. Bbased on the formulation and user selection, the RISDC users are the subset of RIS users, i.e., The user will be RISDC only if its two associated RIS links’ SINR are greater than a certain predetermined threshold, τ. The coverage, RIS user selection, and RISDC user selection threshold are set to be −6.5 dB [21] throughout the simulation. However, these values can be varied based on the scenario and derived results remains the same for all the scenarios. The throughput corresponding to the (Equation 14) is given as follows.
(37)$$T_{\alpha }=\left\{\begin{array}{cc} \;{\left(\frac{1}{U}\sum_{u\in U}\lambda_{u}^{1-\alpha }\right)}^{\frac{1}{1-\alpha }} & ,\alpha >0,\;\alpha \neq 1,\\ \;{\left(\prod_{u\in U}\lambda_{u}\right)}^{\frac{1}{U}} & ,\alpha =1, \end{array}\right.$$
where, α is the fairness parameter, $\lambda_{u}$ is as defined in (Equation 15), and U is the set of users in the system. To study the impact of distributed RIS, the number of RIS-associated to BS are varied. The impact on throughput and coverage by varying user and BS densities. Further, the user density is varied from 20 to 120 users in the considered area (0.04/km^2^) to see the performance of RIS-assisted DC with different user densities. The simulation parameters are specified in Table 2 and a snapshot of simulation settings is shown in Figure 2.

We follow a snapshot-based approach, where the users and blockages are distributed uniformly randomly in the considered area. The simulation is carried out using MATLAB R2020b. We consider throughput for all the users in the system and throughput for non zero rate users in the system. The non zero rate users are defined as the users under coverage by the BSs (its SINR≥−6.5 dB). The variation of throughput with respect to number of RIS in the system is shown in Figure 3. The proposed RISDC system performs better than the benchmark and DC system in Figure 3 with a gain of approximately 52% and 48%, respectively. However, the RISDC performs similarly to the RIS with a gain of 0.56% when we consider throughput of all users. However, if we consider only the users in coverage, the gain due to RISDC is comparably higher than the RIS system which is approximately 2.5%. This is due to the fact that the throughput in (Equation 37) for α=1 is the geometric mean rate. The geometric mean rate is better when all the users’ rates are better. Therefore, in order to observe the gain of RISDC system when compared to the other systems, the throughput for only *users in coverage (non-zero rate users)* are computed and shown in Figure 4. The users with SINR≥−6.5dB are only in coverage and have non-zero rate according to [21]. The significant gain of the RISDC system when compared to other benchmarks (12%), DC(6%), and RIS (2.4%) for systems considering only non zero rate users is observed in Figure 4.

The coverage of the system with varying number of RISs and the user density of $\rho_{user}=60$ for different number of BSs are shown in Figure 5. We evaluated for two mmWave based BSs (2BS) and four mmWave based BSs (4BS) deployed in the same area. The coverage in Figure 5 is computed over $10^{4}$ iterations. Each time, the number of users in coverage are computed and averaged over $10^{4}$ realizations. The coverage with more number of BSs is less than the coverage with less number of BSs. This is because, as the number of BSs in the system increases, the interference also increases which substantially reduces the SINR of the users. In this, the coverage with RIS and RISDC is also less because we consider interference from other BSs for RIS and RISDC users. Further, the throughput performance with varying user densities for different number of BSs in the systems are shown in Figure 6 and Figure 7. Since all the users in the system are considered for computing throughput, the gain that we observe from Figure 6 and Figure 7 is minimal. If we consider throughput of the users in coverage, then a significant gain is achieved with the RISDC system. However, the throughput with 4BSs in Figure 7 is less than the 2BS case as in Figure 6 because it becomes interference limited system when the number of BSs increases.

The users’ rates are arranged in increasing order. We consider evaluating the throughput for α=1 in (Equation 37) for every 10% of users. This is defined as the lowest $m^{th}$ percentile of users. This *m* is varied as every 10% as shown in Figure 8. The same interference limited scenario for increasing number of BSs is observed while computing this worst case users’ throughput also. There is a small gain achieved by RISDC system but the performance is similar when compared to RIS system. Those increase in comparable throughput is maintained even with varying the blockage density in the system as shown in Figure 9. As the average number of blockages ($\mu_{b}$) in the system increases, the gain direct LOS decreases significantly, which in turn reduces the throughput of the system. However, the gain of the proposed RISDC system with respect to the RIS system is 0.56%, 0.94%, 5.19%, and 11.8% for blockage densities of 10, 20, 30, and 40, respectively. The proposed RISDC system is for a α-fair utility function. While the α increases, the fairness to the worst case users’ rate increase at the cost of best case user’s rate. As a result, the throughput of users in coverage starts decreasing as in Figure 10. Further, when α≥2, the benchmark (Ben) and RIS throughput surpasses the DC and RISDC system’s throughput, respectively. As fairness in the system (α) increases, the users’ rate due to DC in benchmark and RIS system saturates as there is only limited resource available. Hence, RIS and RISDC system improves the coverage and data rate of the user instead of throughput of the system when α≥2. This improvement in data rate of the proposed RISDC system leads to small gain in terms of throughput but the gain does not change considerably. However, *the proposed α-fair scheduling for RIS-assisted DC outperforms other benchmark and DC system in terms of throughput for various BS, user, and blockage densities.*

## 6. Conclusions and Future Work

The RIS-assisted DC system is proposed by utilizing the RIS resource and radio resource from two different RIS and BS, respectively. The joint resource allocation for RIS-assisted DC system is formulated as an optimization problem and the user scheduling time fraction is derived for an α-fair scheduler. Further, a heuristic for solving given RIS-assisted DC problem with the derived optimal user scheduling time fractions is presented. The exhaustive simulation results of throughput are presented for different user, BS, and blocker densities. The proposed system provides significant throughput gain of 50% and 48% in certain scenarios when compared to the existing benchmark and DC systems. It is also shown that the coverage with increased number of BSs in the system decreases and saturates beyond certain RIS density. For RIS to be advantageous as a standalone and RIS-assisted DC system, the RIS deployment should be carefully designed for any BS. The proposed heuristic categorizes the users based on the available channel parameters. In the future, we consider evaluating the performance of various RIS-assisted wireless systems during uplink and with imperfect phase estimation.

## Figures and Tables

**Figure 1 sensors-22-05755-f001:**
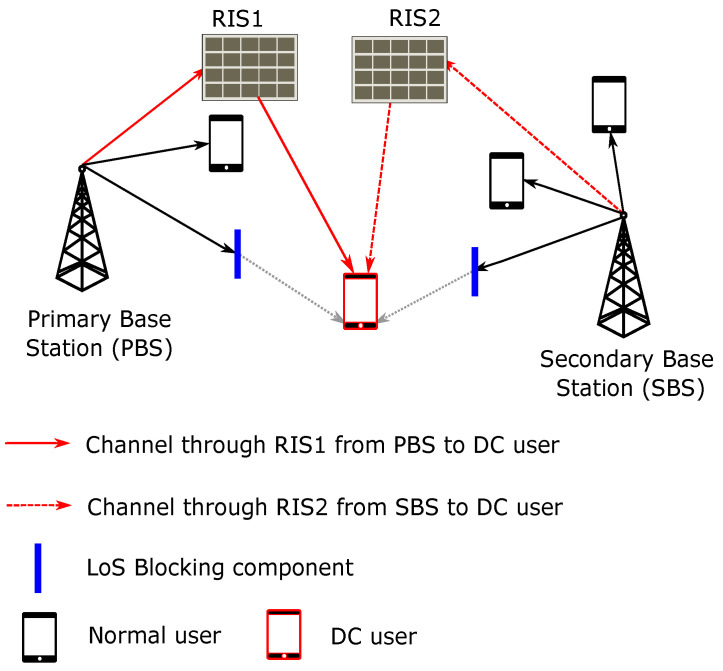
System Model of RIS-assisted DC.

**Figure 2 sensors-22-05755-f002:**
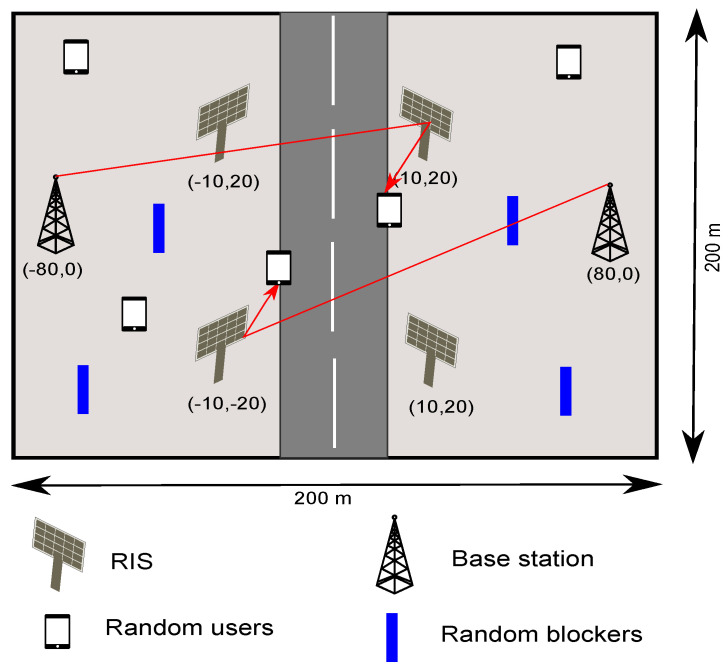
Simulation settings.

**Figure 3 sensors-22-05755-f003:**
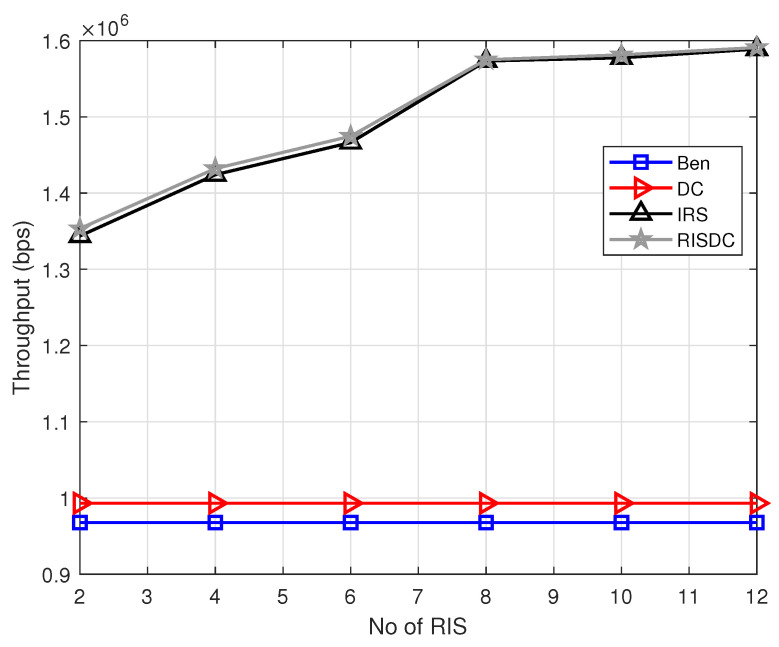
Variation of throughput with respect to number of RISs when $\rho_{bs}=2$, $\rho_{user}=60$, and $\mu_{b}=10$.

**Figure 4 sensors-22-05755-f004:**
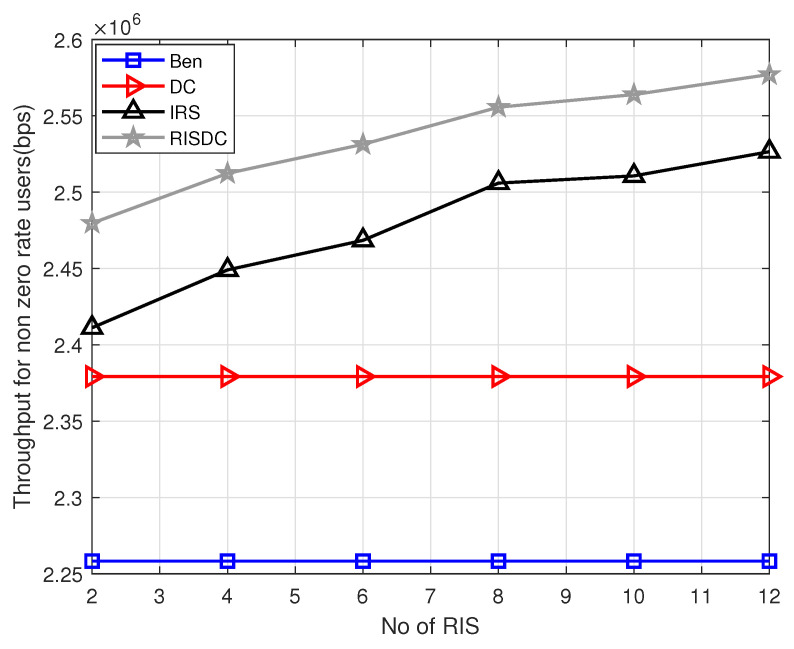
Variation of throughput of non-zero rate users (users only in coverage) with respect to number of RISs when $\rho_{bs}=2$, $\rho_{user}=60$, and $\mu_{b}=10$.

**Figure 5 sensors-22-05755-f005:**
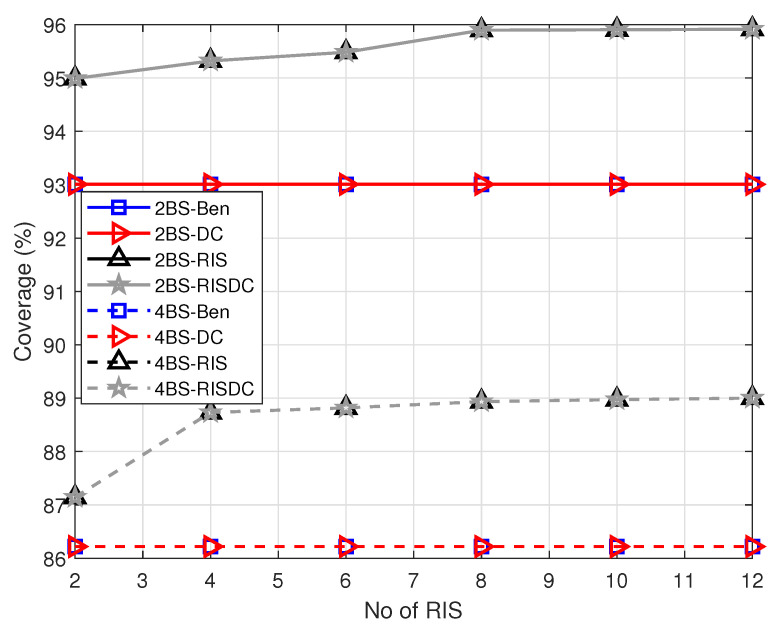
Variation of coverage with respect to number of RISs when $\rho_{bs}=\left[2,4\right]$, $\rho_{user}=60$, α=1, and $\mu_{b}=10$.

**Figure 6 sensors-22-05755-f006:**
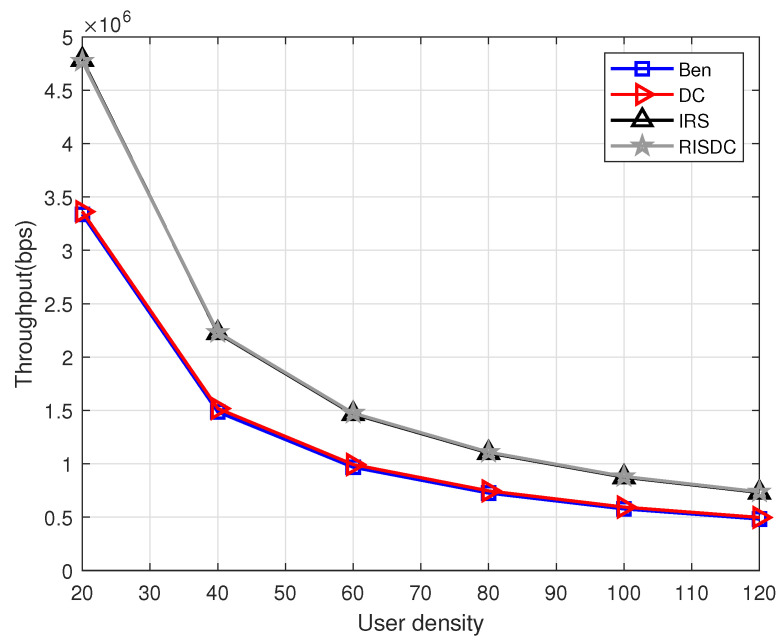
Variation of throughput with respect to number of users when $\rho_{bs}=2$, $\rho_{ris}=6$, α=1, and $\mu_{b}=10$.

**Figure 7 sensors-22-05755-f007:**
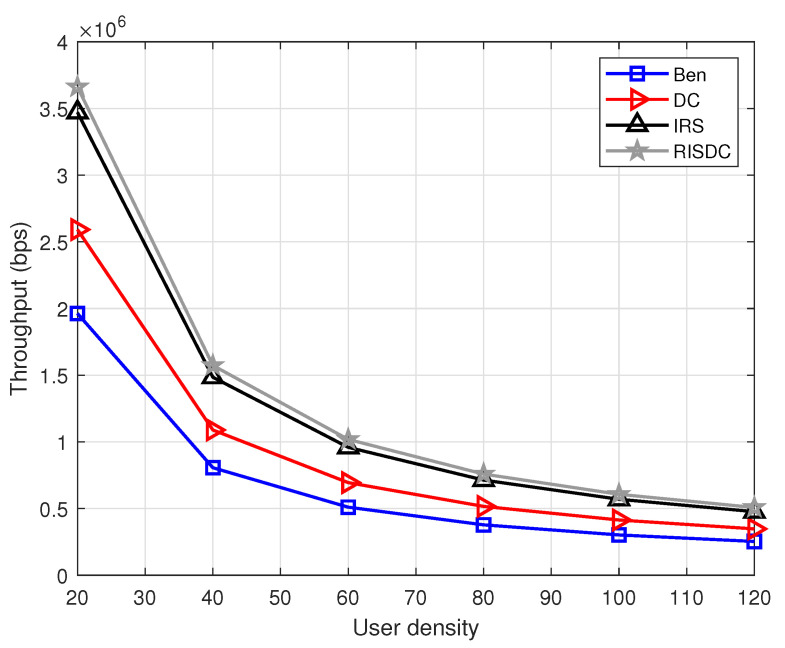
Variation of throughput with respect to number of users when $\rho_{bs}=4$, $\rho_{ris}=6$, α=1, and $\mu_{b}=10$.

**Figure 8 sensors-22-05755-f008:**
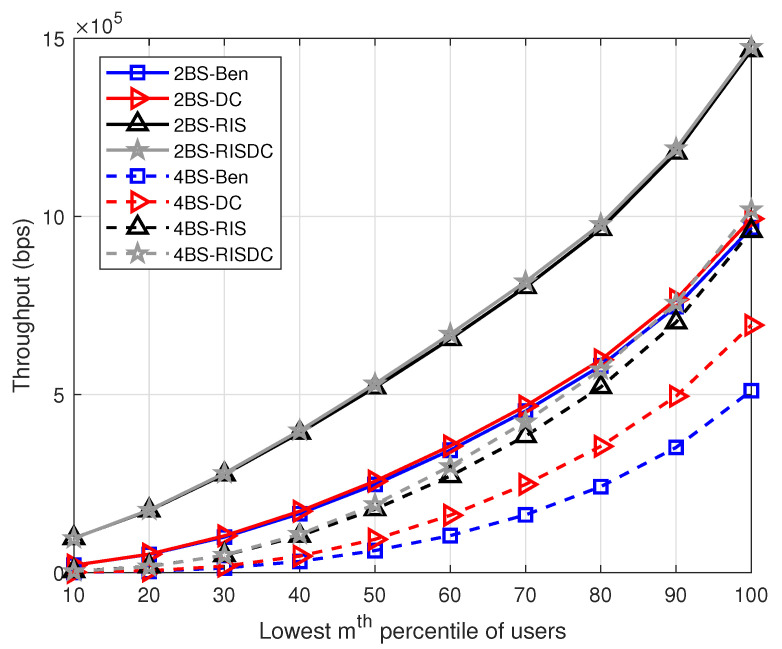
Variation of throughput with respect to lowest $m^{th}$ percentile of users when $\rho_{bs}=\left[2,4\right]$, $\rho_{user}=60$, α=1, and $\mu_{b}=10$.

**Figure 9 sensors-22-05755-f009:**
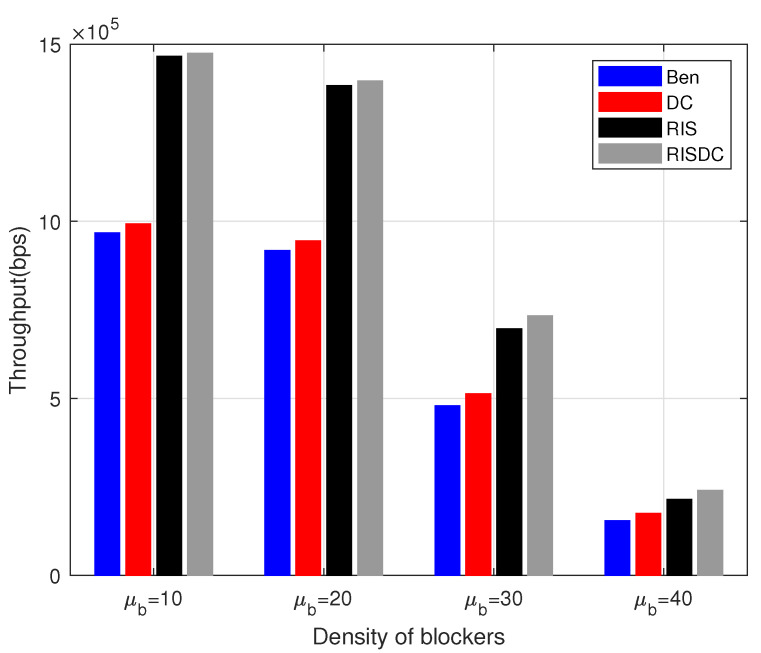
Variation of throughput with respect to blockage density $\mu_{b}$ when $\rho_{bs}=2$, $\rho_{user}=60$, $\rho_{ris}=6$, and α=1.

**Figure 10 sensors-22-05755-f010:**
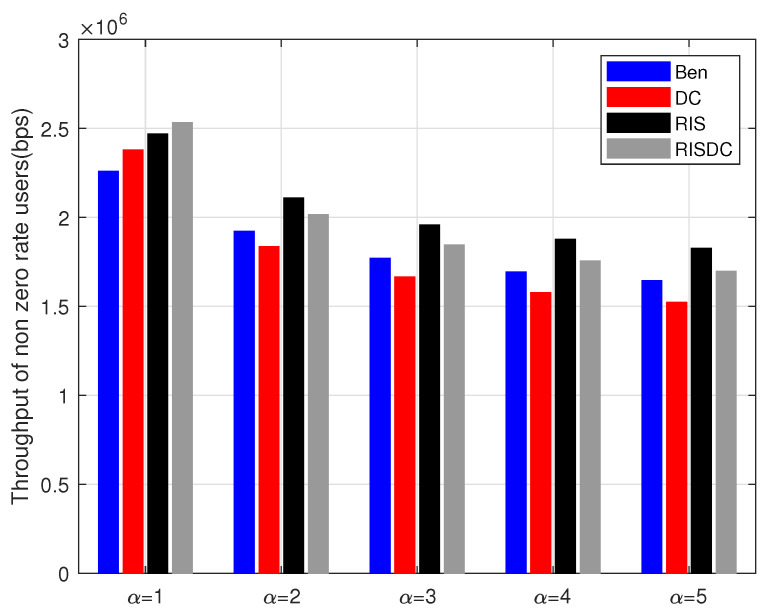
Variation of non zero rate users’ throughput for different αs when $\rho_{bs}=2$, $\rho_{user}=60$, $\rho_{ris}=6$, and $\mu_{b}=10$.

**Table 1 sensors-22-05755-t001:** Mathematical notations.

$h_{u,j}^{0}$	Direct channel gain at user *u* from BS *j*
${\hat{h}}_{u,j}^{r}$	Channel gain at user *u* from BS *j* via RIS *r*
$l_{u,j}^{0}$	Direct link rate of user *u* from BS *j*
$l_{u,j}^{r}$	Link rate of user *u* from BS *j* via RIS *r*
$x_{u,j}^{0}$	Binary user association variable of user
	*u* with BS *j* without RIS
$x_{u,j}^{r}$	Binary user association variable of user
	*u* with BS *j* via RIS
α	Fairness parameter for the α-Fair scheduler
$\delta_{u,j}^{0}$	User scheduling time fraction for user *u* by BS *j* without RIS
$\delta_{u,j}^{r}$	User scheduling time fraction for user *u* by BS *j* via RIS *r*
Γ(.)	Spectral efficiency in bits/symbol
$\mu_{b}$	Blockage density
$\omega_{u,j}^{0}$	DL received SINR of user *u* from a BS *j*
$\omega_{u,j}^{r}$	DL received SINR of user *u* from a BS *j* via RIS *r*
$\rho_{bs}$	Number of BSs
$\rho_{user}$	Number of users
Λ	Utility function
$d_{u,j}^{r}$	Binary association variable for DC user
$\lambda_{u}$	Data rate of user *u*
$\beta_{r,n}$	Reflection amplitude of $n^{th}$ antenna element of $r^{th}$ RIS
$\theta_{r,n}$	Phase shift of of $n^{th}$ antenna element of $r^{th}$ RIS
$T_{\alpha }$	Throughput

**Table 2 sensors-22-05755-t002:** Simulation Parameters.

$f_{c}$	28 GHz
Penetration loss ($\upsilon_{sc}$)	20 dB for NLOS path
Loss due to shadowing (ρ)	Standard deviation of 4 dB
P${}_{j}$	30 dBm
PL(d)	Urban micro [23]
*C*	99
Subchannel Bandwidth	720 KHz
SC${}_{OFDM}$	12
SY${}_{OFDM}$	14
T${}_{Subframe}$	0.25 ms

## Data Availability

Not applicable.

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
