# Peer review of "Resource Allocation for Reconfigurable Intelligent Surface Assisted Dual Connectivity"

_sensors, 2022, doi:10.3390/s22155755_

Round 1

Reviewer 1 Report

This paper propose the reconfigurable intelligent surface (RIS) assisted dual connectivity (DC) system by utilizing resources from two base stations (BSs) in proximity via different RISs, which can effectively improve the per-user throughput of the users. The proposed system has a reference role in improving the performance of next generation 6G wireless systems. The idea is novel and interesting. Therefore, this paper can be accepted with major revisions. The points need to be addressed are listed as follows.

1.     Can you further refine the first section of your paper?

2.     In addition to coverage and α-fair throughput, is there other performance metrics in RIS assisted DC system?

3.     Compared with the previous system, the RIS assisted DC system has achieved a great throughput gain. Can you briefly analyze the cost in resource allocation.

4.     The simulation model in Section V is relatively simple. Can the number of base stations be increased considering the boundary conditions.

5.     In the conclusion part, please put forward some deficiencies of the current research.

Author Response

Resource Allocation For Reconfigurable Intelligent Surface Assisted Dual Connectivity

sensors-1818530

Yoghitha Ramamoorthi * ,Masashi Iwabuchi, Tomoki Murakami, Tomoaki Ogawa, and Yasushi Takatori

Dear Reviewer 1

Thank you for your kind comments and suggestions. Please find our responses below.

All reviewers’ comments in this document are written in black and their responses are written in red. The text in blue color shows the updated portion/portion of the revised manuscript.

General Comment

This paper propose the reconfigurable intelligent surface (RIS) assisted dual connectivity (DC) system by utilizing resources from two base stations (BSs) in proximity via different RISs, which can effectively improve the per-user throughput of the users. The proposed system has a reference role in improving the performance of next generation 6G wireless systems. The idea is novel and interesting. Therefore, this paper can be accepted with major revisions. The points need to be addressed are listed as follows.

Response : Thank you for your kind comments. Please find below the responses to your comments. We have made minor revisions to the manuscript based on the other reviews. We hope that you will be satisfied with the revised version of this manuscript.

Response to Reviewer1’s Comments

    Point 1: Can you further refine the first section of your paper?

Response 1: Thank you for your kind comment. We have presented the first section in the following format. Introduction to RIS and some of the RIS related worksfrom the resource allocation and scheduling perspective. As a next step, the dual connectivity (DC) is introcuced and RIs along with multi-transmit entities are presented. Finally, the flaws in the existing literature along with the contributions made in this manuscript are discussed. However, based on your review, we have refined the Section. 1 with its articulation.

    Point 2: In addition to coverage and α-fair throughput, is there other performance metrics in RIS assisted DC system?

Response 2: Thank you for your kind comment. There are other performance metrics like spectral efficiency, energy efficiency, etc, that can be presented for the same RIS assisted DC system. However, we focused on system-level parameters such as coverage and α-fair throughput as our performance metrics. Please note that we have presented system level simulation results after optimal scheduling. Further, we formulated our optimization problem with the objective of maximizing system throughput considering all users. Thus, based on the formulated problem, it would be better to explain the performance metric in terms of coverage and α-fair throughput.

       Point 3: Compared with the previous system, the RIS assisted DC system has achieved a great throughput gain. Can you briefly analyze the cost in resource allocation.

    Response 3: Thank you for your kind comment. Yes, we agree, the RIS assisted DC system achieved a great throughput gai when compared to benchmark and dual connectivity (DC) system without RIS. We discussed the complexity of the heuristic that includes resource allocation and scheduling as below

  • On Section. 4, page. 10, below Algorithm. 1, line. 18, “The complexity of the heuristic is in the order of 2U, where U is the total number of users in the system. The heuristic is divided into two main steps. The computation of SINR and the selection of normal, DC, and RIS assisted DC users at the first step. The latter step continues with computation of optimal user scheduling time fraction and the resultant data rate”.

According to the above heuristic, this scheduling and resource allocation of RIS assisted DC and RIS system have the complexity of 2U, whereas, the benchmark and DC system without RIS have the complexity of U.

    Point 4: The simulation model in Section V is relatively simple. Can the number of base stations be increased considering the boundary conditions.

    Response 1: Thank you for your kind comment. In the simulation settings Figure. 2, we have shown only 2 BSs just for understanding. However, we presented coverage and throughput for different number of BSs (ρbs=2,4) in our simulation as in Figure. 5, 6, 7, and 8. 

Point 5: In the conclusion part, please put forward some deficiencies of the current research.

Response 1: Thank you for your kind comments.

  • On page 17, Section 6, “For RIS to be advantageous as standalone and RIS assisted DC system, the RIS deployment should be carefully designed for any BS. The proposed heuristic categorizes the users based on the available channel parameters. In the future, we consider evaluating the performance of various RIS assisted wireless systems during uplink and with imperfect phase estimation.”

Even though RIS assisted DC system has greater gains when compared to benchmark and DC system without RIS. The planning and placement of RIS should be proper in order to achieve these gains. Otherwise, the gain might not be huge when compared to RIS system.

We have made minor revisions to the manuscript based on the other reviews. We hope that you will be satisfied with the revised version of this manuscript.

Reviewer 2 Report

Simulations show the usefulness of resource allocation techniques for RISDC.

In order to clarify the reliability of this proposed method, please consider experimental evaluation in the future.

There is a part where the proofreading of the text is insufficient.

Author Response

Resource Allocation For Reconfigurable Intelligent Surface Assisted Dual Connectivity

sensors-1818530

Yoghitha Ramamoorthi * ,Masashi Iwabuchi, Tomoki Murakami, Tomoaki Ogawa, and Yasushi Takatori

Dear Reviewer 2

Thank you for your kind comments and suggestions. Please find our responses below.

All reviewers’ comments in this document are written in black and their responses are written in red. The text in blue color shows the updated portion of the revised manuscript.

Response to Reviewer2’s Comments

Point 1: Simulations show the usefulness of resource allocation techniques for RISDC.

Response 1: Thank you for your kind comment.

Point 2: In order to clarify the reliability of this proposed method, please consider experimental evaluation in the future.

Response 1: Thank you for your kind comment. We agree that the proposed method leverages the advantages of RIS and dual connectivity (DC) and improves the throughput and coverage. We plan We will consider the experimental evaluation in future.

Point 3: There is a part where the proofreading of the text is insufficient.

    Response 1: Thank you for your kind. We will properly proofread the manuscript completely while resubmission.

We have made minor revisions to the manuscript based on the other reviews. We hope that you will be satisfied with the revised version of this manuscript.

Reviewer 3 Report

This manuscript investigates RIS-assisted dual connectivity (DC) systems to improve the throughput of users. Overall, the manuscript is well written and easy to follow. The detailed comments/suggestions are as follows.  

1) The authors claimed that this is the first work incorporating RIS with DC. However, recent studies addressed this issue with UAVs carrying RIS-assisted DC systems. Please check the literature carefully. With this said, please revise the Introduction part to highlight differences, motivations, and contributions more clearly.  

2) Please cite equations properly as I believe that many equations were adopted from other works. 

3) How about the imperfect phase estimation of RIS? It seems the authors did not mention this issue in the analysis.  

4) How many RIS elements are considered in the manuscript? 

5) Please use citations for considered systems in Section 5. 

Minor comments: 

1) Line 72: “On is primary BS” →  “One is primary BS”

2) Quality of figures, e.g., figs 1 and 2, should be improved.

Author Response

Resource Allocation For Reconfigurable Intelligent Surface Assisted Dual Connectivity

sensors-1818530

Yoghitha Ramamoorthi * ,Masashi Iwabuchi, Tomoki Murakami, Tomoaki Ogawa, and Yasushi Takatori

Dear Reviewer 3

Thank you for your kind comments and suggestions. Please find our responses below.

All reviewers’ comments in this document are written in black and their responses are written in red. The text in blue color shows the updated portion of the revised manuscript.

General Comment

This manuscript investigates RIS-assisted dual connectivity (DC) systems to improve the throughput of users. Overall, the manuscript is well written and easy to follow. The detailed comments/suggestions are as follows. 

Response: Thank you for your kind comments. Please find below the responses to your comments. We have made minor revisions to the manuscript based on the other reviews. We hope that you will be satisfied with the revised version of this manuscript.

Response to Reviewer3’s Comments

    Point 1: The authors claimed that this is the first work incorporating RIS with DC. However, recent studies addressed this issue with UAVs carrying RIS-assisted DC systems. Please check the literature carefully. With this said, please revise the Introduction part to highlight differences, motivations, and contributions more clearly. 

Response 1: Thank you for your kind comment. There are some works addressing UAV carrying RIS systems. However, to our knowledge, this is the first work that combines RIS with DC and the optimal α-fair scheduling (derived) for RIS assisted DC system. Based on your review, the Introduction, motivation, and the contribution part is slightly modified and the details are given below.

We have presented the first section in the following format. Introduction to RIS and some of the RIS related worksfrom the resource allocation and scheduling perspective. As a next step, the dual connectivity (DC) is introcuced and RIs along with multi-transmit entities are presented. Finally, the flaws in the existing literature along with the contributions made in this manuscript are discussed. However, based on your review, we have refined the Section. 1 with its articulation.

    Point 2: Please cite equations properly as I believe that many equations were adopted from other works

Response 2: Thank you for your kind comment. All the system model equations like channel gain, SINR, link rate, scheduling, utility function, etc, are adopted from the basic works. The references are now in the appropriate places in the revised manuscript. 

       Point 3: How about the imperfect phase estimation of RIS? It seems the authors did not mention this issue in the analysis.

    Response 3: Thank you for your kind comment. We did not consider phase estimation in this analysis. We focused on the optimal user scheduling  and rewsource allocation of RIS assisted DC system given the channel gain parameters. However, the phase estimation with this proposed system is an interesting analysis and we consider it in our future work.

  • On page 17, Section 6, the last few lines are modified as “For RIS to be advantageous as standalone and RIS assisted DC system, the RIS deployment should be carefully designed for any BS. The proposed heuristic categorizes the users based on the available channel parameters. In the future, we consider evaluating the performance of various RIS assisted wireless systems during uplink and with imperfect phase estimation.”

    Point 4: How many RIS elements are considered in the manuscript?

    Response 4: We considered 40 elements in each RIS.

Point 5: Please use citations for considered systems in Section 5.

Response 5: Thank you for your kind comment. We have included the citiations for benchmark and DC systems in Section. 5.

Point 6: Line 72: “On is primary BS” →  “One is primary BS”

Response 6: Thank you for your kind comments. The given mistake is corrected.

Point 7: Quality of figures, e.g., figs 1 and 2, should be improved.

Response 7: Thank you for your kind comment. We have changed the resolution and size of the above said figures in the revised manuscript.

We have made minor revisions to the manuscript based on the other reviews. We hope that you will be satisfied with the revised version of this manuscript.

Round 2

Reviewer 1 Report

I have no other review comments.

Reviewer 3 Report

I have no further comments for this manuscript. It's ready for publication.